# Continuation of Radial Positive Definite Functions and Their Characterization

**Fethi Bouzeffour** 

Department of Mathematics, College of Sciences, King Saud University, P.O. Box 2455, Riyadh 11451, Saudi Arabia; fbouzaffour@ksu.edu.sa

**Abstract:** This paper delves into the extension and characterization of radial positive definite functions into non-integer dimensions. We provide a thorough investigation by employing the Riemann–Liouville fractional integral and fractional Caputo derivatives, enabling a comprehensive understanding of these functions. Additionally, we introduce a secondary characterization based on the Bernstein characterization of completely monotone functions. The practical significance of our study is showcased through an examination of the positivity of the fundamental solution of the space-fractional Bessel diffusion equation, highlighting the real-world applicability of the developed concepts. Through this work, we contribute to the broader understanding of radial positive definite functions and their utility in diverse mathematical and applied contexts.

**Keywords:** positive definite functions; completely monotone functions; fractional integral and derivative; fractional diffusion equation



## 1. Introduction

Radial positive definite functions are fundamental in various mathematical disciplines, such as functional analysis, probability theory, signal processing, and more [1–3]. They exhibit essential properties that make them indispensable in diverse applications, including defining positive definite kernels in machine learning, stochastic processes in probability theory, and generating interpolating functions in signal processing [3].

Traditionally, the study of radial positive definite functions has been confined to Euclidean spaces with integer dimensions. However, there has been a growing interest in extending these functions beyond integer dimensions to investigate their behavior in more general spaces, such as fractional and non-integer dimensions. Such extensions provide a deeper understanding of the underlying mathematical structures and open up new avenues for practical applications in areas such as image processing, geostatistics, and fractional calculus.

The works of Cholewinski et al. [4] and Chebli [5] have made significant contributions to this field by exploring the continuation of radial positive definite functions associated with Bessel operators of arbitrary order and a family of singular regular differential operators. Additionally, Trimeche [6] has investigated the extension of these functions in the context of Bessel and Jacobi operators, enhancing our understanding of their behavior in different operator settings.

Fractional calculus, which deals with derivatives and integrals of non-integer order, provides a powerful framework for analyzing functions with non-local and long-range dependencies. This makes it particularly well-suited for the study of radial positive definite functions beyond integer dimensions. Numerous researchers have employed fractional calculus to explore the properties and behavior of such functions, leading to insights into their fractional derivatives and fractional integrals [1,2,7–12].

The primary objective of this study is to delve into the continuation of radial positive definite functions and their characterization using fractional derivatives. Building upon the

foundational works of Cholewinski et al. and Chebli, we aim to establish a comprehensive understanding of these functions in non-integer dimensions. Furthermore, we introduce an alternative characterization based on completely monotone functions, which have significant properties and find extensive applications in various mathematical areas [13,14].

To demonstrate the practical relevance of our findings, we explore the application of these extended positive definite functions in the context of the space fractional diffusion equation. By utilizing the obtained characterization, we investigate the positivity of the fundamental solution of this equation, shedding light on the practical implications and potential applications of the continued radial positive definite functions in solving real-world problems.

The paper is structured as follows: In Section 2, we provide an overview and introduce the notations and key facts related to the Fourier–Bessel transform and the Delsarte translation, essential tools for the subsequent analyses.

Section 3 delves into the concept of radial positive definite functions, with a particular focus on the renowned Bochner and Schoenberg theorems. Additionally, we explore the continuation of radial positive definite functions in the context of fractional dimensions, showcasing their behavior beyond integer dimensions.

In Section 4, we present a detailed characterization of the representation of functions on the interval $(0, \infty)$ as Fourier–Bessel transforms using Riemann–Liouville integrals and Caputo fractional derivatives. These characterizations reveal the underlying structure of these functions and provide insights into their fractional behavior.

Section 5 focuses on the characterization of the class $\mathcal{P}_\kappa$ by leveraging Bernstein's theorem for completely monotone functions. This alternative characterization offers a complementary perspective and further enriches our understanding of radial positive definite functions.

Finally, in the last section, we demonstrate the positivity of the fundamental solution of the space-fractional Bessel diffusion equation, showcasing the practical implications of our study and highlighting the utility of extended radial positive definite functions in solving real-world problems.

## 2. Preliminaries

This section serves as an introduction to the notations and key facts related to the Fourier–Bessel transform and the Delsarte translation. These concepts are essential for understanding the subsequent analysis and characterization of radial positive definite functions beyond integer dimensions.

First, we introduce the normalized Bessel function, denoted as $\mathscr{J}_\kappa(x)$, which plays a fundamental role in the sequel. The normalized Bessel function is defined by

$$\mathscr{J}_\kappa(x) := \Gamma(\kappa + 1)\,(2/x)^\kappa\, J_\kappa(x), \quad \kappa > -1,$$

where $\Gamma(\cdot)$ is the Gamma function [15] and $J_\kappa(\cdot)$ is the Bessel function, see [(10.16.9)] [16]. Then,

$$\mathscr{J}_\kappa(x) = \sum_{k=0}^{\infty} \frac{(-\frac{1}{4}x^2)^k}{(\kappa+1)_k\, k!} = {}_0F_1\left(\begin{matrix} - \\ \kappa+1 \end{matrix}; -\frac{1}{4}x^2\right) \qquad (\kappa > -1).$$

Here, $(\kappa+1)_k$ denotes the Pochhammer symbol, and ${}_0F_1(\cdot)$ represents the confluent hypergeometric function [15]. The normalized Bessel function emerges as the unique solution of the eigenvalue problem associated with the Bessel equation. Specifically, the functions $x \to \mathscr{J}_\kappa(\lambda x)$ is the unique solution of the eigenvalue problem [(10.13.5)] [16]

$$\begin{cases} \mathscr{B}_\kappa u(x) = -\lambda^2 u(x), \\ u(0) = 1, \quad u'(0) = 0. \end{cases}$$

where $\mathscr{B}_\kappa$ is the Bessel operator

$$\mathscr{B}_\kappa := \frac{d^2}{dx^2} + \frac{2\kappa + 1}{x}\frac{d}{dx}, \quad \kappa \geq -1/2.$$

The function $\mathscr{J}_\kappa(\cdot)$ is an even entire analytic function, and we have the simple special cases:

$$\mathscr{J}_{-1/2}(x) = \cos x, \quad \mathscr{J}_{1/2}(x) = \frac{\sin x}{x}.$$

By ([§7.21] [16]), we have the asymptotic expansion of the normalized Bessel function

$$\mathscr{J}_\kappa(x) = \cos(x - \frac{\kappa\pi}{2} - \frac{\pi}{4}) + O(\frac{1}{x}), \quad x \to \infty, \quad \kappa > -\frac{1}{2}. \tag{1}$$

We denote by $L_\kappa^p(0, \infty)$ $(1 \leq p)$, the Lebesgue space associated with the measure

$$\sigma_\kappa(dx) = \frac{x^{2\kappa+1}}{2^{\kappa+1}\Gamma(\kappa+1)}dx$$

and by $\|f\|_{\kappa,p}$ the usual norm given by

$$\|f\|_{\kappa,p} = \left(\int_0^\infty |f(x)|^p\, \sigma_\kappa(dx)\right)^{1/p}.$$

The Fourier–Bessel transform $\mathscr{F}_\kappa f$ of $f \in L_\kappa^1(0, \infty)$ is defined as:

$$\mathscr{F}_\kappa f(x) = \int_0^\infty f(t)\, \mathscr{J}_\kappa(tx)\sigma_\kappa(dx), \quad \kappa \geq -1/2. \tag{2}$$

This integral transform plays a similar role as the Fourier transform in the Euclidean one. In particular, it can be extended to an isometry of $L_\kappa^2(0, \infty)$, and for any $f \in L_\kappa^1(0, \infty) \cap L_\kappa^2(0, \infty)$, we have

$$\int_0^\infty |f(x)|^2\, \sigma_\kappa(dx) = \int_0^\infty |\mathscr{F}_\kappa f(t)|^2\, \sigma_\kappa(dt) \tag{3}$$

and its inverse is given by

$$f(x) = \int_0^\infty \mathscr{F}_\kappa f(t)\, \mathscr{J}_\kappa(tx)\, \sigma_\kappa(dt). \tag{4}$$

Next, we discuss the generalized translation operator associated with the Bessel operator. This operator is denoted as $\tau_\kappa^x$ and acts on functions $f \in L_\kappa^1(0, \infty)$ as follows [§3.4.1] [17]:

$$\tau_\kappa^x f(y) = \begin{cases} \int_0^\pi f(\sqrt{x^2 + y^2 + 2xy\cos\theta})\sin^{2\kappa}\theta\, d\theta, & \text{if } \kappa > -1/2, \\ \frac{1}{2}(f(x+y) + f(x-y)), & \text{if } \kappa = -1/2. \end{cases} \tag{5}$$

With the help of this translation operator, one defines the convolution of $f \in L_\kappa^1(0, \infty)$ and $g \in L_\kappa^p(0, \infty)$ for $p \in [1, \infty)$ as the element $f *_\kappa g$ of $L_\kappa^p(0, \infty)$ given by

$$(f *_\kappa g)(x) := \int_0^\infty (\tau_\kappa^x f)(y)\, g(y)\sigma_\kappa(dy), \quad \kappa \geq -1/2. \tag{6}$$

The following properties are obvious:

- $\mathscr{F}_\kappa(\tau_\kappa^x f)(t) = \mathscr{J}_\kappa(xt)\mathscr{F}_\kappa f(t);$
- $\mathscr{F}_\kappa(f *_\kappa g)(x) = \mathscr{F}_\kappa f(x)\mathscr{F}_\kappa g(x).$

### 3. Continuation of Radial Positive Definite Functions

In this section, we will delve into the concept of radial positive definite functions, with a particular focus on the renowned Bochner and Schoenberg theorems. Additionally, we explore the continuation of radial definite positive functions.

A complex-valued function $f$ defined on $\mathbb{R}^n$ is said to be definite positive and belongs to the class $\Phi(\mathbb{R}^n)$ if it is continuous at the origin and the matrix [13,14,18]

$$\left(f(x_i - x_j)\right)_{i,j=1}^N$$

is non-negative definite for all finite systems of points $x_1, x_2, \ldots, x_N \in \mathbb{R}^n$.

The classical Bochner's theorem [18] provides a fundamental characterization of the class $\Phi(\mathbb{R}^n)$. Specifically, a function $f$ belongs to $\Phi(\mathbb{R}^n)$ if and only if it can be expressed as the Fourier transform of a finite non-negative Borel measure $\mu$ on $\mathbb{R}^n$ as:

$$f(x) = \int_{\mathbb{R}^n} e^{-i\langle x,t \rangle} \mu(dt), \tag{7}$$

where $x, t \in \mathbb{R}^n$, and $\langle x, t \rangle$ denotes the inner product between $x$ and $t$.

In the context of radial functions, let us recall that a function $f$ defined on $\mathbb{R}^n$ is considered radial if there exists an even function $f_0$ defined on $\mathbb{R}$ such that $f(x) = f_0(\|x\|)$ for all $x \in \mathbb{R}^n$, where $\|\cdot\|$ denotes the Euclidean norm on $\mathbb{R}^n$. In other words, the value of the radial function $f(x)$ depends solely on the magnitude or norm of the vector $x$. By considering the properties of radial functions, we obtain the following expression for the Fourier transform:

$$\mathcal{F}f(\xi) = \frac{1}{(2\pi)^n} \int_{\mathbb{R}^n} f(x) e^{-ix \cdot \xi} dx, \quad \xi \in \mathbb{R}^n$$
$$= \frac{2\pi^{n/2}}{\Gamma(n/2)} \int_0^\infty f_0(r) \mathscr{J}_{n/2-1}(\|\xi\|r) r^{n-1} dr.$$

A function $f : [0, \infty) \to \mathbb{R}$ is considered a radial positive definite function of the class $\Phi_n$, if the function $f(\|\cdot\|)$ belongs to the class $\Phi(\mathbb{R}^n)$. The class $\Phi_n$ is characterized by the Schoenberg theorem, which is referenced as [14].

**Theorem 1** ([4]). *Function $f(\cdot)$ belongs to the class $\Phi_n$ if and only if*

$$f(x) = \int_0^\infty \mathscr{J}_{n/2-1}(tx) \mu(dt), \tag{8}$$

*where $\mu$ is a non-negative finite Borel measure on $[0, \infty)$.*

In [4], the authors introduced an innovative class of positive definite functions that relies on the generalized translation of Delsarte $\tau_\kappa^x$, as in Equation (5).

**Definition 1** ([4]). $\mathscr{P}_\kappa$ *is the set of continuous functions $f : [0; \infty) \to \mathbb{R}$ such that the matrix*

$$\left(\tau_\kappa^{x_i} f(x_j)\right)_{i,j=1}^N$$

*is non-negative definite for all finite systems of points $x_1, x_2, \ldots, x_N \in [0, \infty)$.*

Note that the discrete condition in the aforementioned definition implies the continuity of its integral counterpart function $f(x)$ on $[0, \infty)$. A bounded function $f(x)$ is said to be positive definite if for every $\varphi \in D_0(\mathbb{R})$ (where $D_0(\mathbb{R})$ denotes the space of even $C^\infty$ functions on $\mathbb{R}$ with compact support), the following inequality holds:

$$\int_0^\infty \int_0^\infty \tau_\kappa^t f(x) \varphi(x) \varphi(t) \sigma_\kappa(x) \sigma_\kappa(t) \geq 0. \tag{9}$$

For more details, see [4,5].

In [4], we found the following characterization of positive definite functions for the Bessel operator, see also [6].

**Theorem 2** ([4]). *A continuous function $f(x)$ is a bounded positive definite function for the Bessel operator $\mathscr{B}_\nu$ if and only if there exists a non-negative finite Borel measure $\mu$ on $[0, \infty)$ such that for every $x \geq 0$,*

$$f(x) = \int_0^\infty \mathscr{J}_\kappa(tx)\mu(dt). \tag{10}$$

In the following proposition, we will show that, when $\kappa$ is half-integer ($\kappa = n/2 - 1$), the class $\mathscr{P}_\kappa$ coincides with $\Phi_n$.

**Proposition 1.** *For every $n \in \mathbb{N}$, we have $\mathscr{P}_{n/2-1} = \Phi_n$.*

**Proof.** Let $f$ be an integrable radial function $f(x) = f_0(\|x\|)$ on $\mathbb{R}^n$. Since the Lebesgue measure is invariant under the orthogonal transformation, the function

$$x \to \int_{\mathbb{R}^n} f(x - y)dy$$

is a radial function. Therefore,

$$\int_{\mathbb{R}^n} f(x - y)dy = \int_{\mathbb{R}^n} f(\|x\|e_1 - y)dy,$$

where $e_1 = (1, 0, \ldots, 0)$. It follows

$$\int_{\mathbb{R}^n} f(x - y)dy = \int_{\mathbb{R}} \int_{\mathbb{R}^{n-1}} f_0(\sqrt{(\|x\| + p)^2 + \|y\|^2})dydp$$

$$= \omega_{n-2} \int_{\mathbb{R}} \int_0^\infty f_0(\sqrt{(\|x\| + p)^2 + \rho^2})\rho^{n-2}d\rho\, dp.$$

where

$$\omega_{n-1} = \frac{2\pi^{n/2}}{\Gamma(n/2)}.$$

Making the substitution $p = t\cos\theta, \rho = t\sin\theta$

$$\int_{\mathbb{R}^n} f(x - y)dy = \omega_{n-2} \int_0^\infty \int_0^\pi f_0(\sqrt{\|x\|^2 + t^2 + 2t\|x\|\cos\theta})\sin^{n-2}(\theta)\,t^{n-1}\,dt$$

$$= \int_0^\infty \tau_{n/2-1}^{\|x\|} f_0(r)\sigma_{n/2-1}(dr).$$

The result follows from Equation (9) and Proposition 6.4 in [3]. □

## 4. Characterization of Positive Definite Functions via Caputo Fractional Derivatives

In this section, we present a characterization of the representation of functions on the interval $(0, \infty)$ as Fourier–Bessel transforms using the Riemann–Liouville integral and the Caputo fractional derivative.

To begin, let us recall the definitions of left-sided and right-sided fractional Riemann–Liouville integrals of order $\alpha$ [12].

For $\mathrm{Re}(\alpha) > 0$, the left-sided fractional Riemann–Liouville integral of order $\alpha$ is given by:

$$I_+^\alpha\{f(t); x\} := \frac{1}{\Gamma(\alpha)} \int_0^x \frac{f(t)dt}{(x-t)^{1-\alpha}} \quad (x > 0). \tag{11}$$

Similarly, the right-sided fractional Riemann–Liouville integral of order $\alpha$ is defined as:

$$I_-^\alpha \{f(t); x\} := \frac{1}{\Gamma(\alpha)} \int_x^\infty \frac{f(t)dt}{(t-x)^{1-\alpha}} \quad (x \geq 0). \tag{12}$$

These integrals involve the gamma function $\Gamma(\alpha)$, which is defined as the integral:

$$\Gamma(\alpha) = \int_0^\infty t^{\alpha-1} e^{-t} dt \quad \alpha > 0. \tag{13}$$

When $\alpha = n$ is a positive integer, we have

$$I_+^n \{f(t); x\} = \frac{1}{(n-1)!} \int_0^x f(t)(x-t)^{n-1} dt$$
$$= \int_0^x dx_1 \int_0^{x_1} dx_2 \cdots \int_0^{x_{n-1}} f(x_n) dx_n.$$

The following semigroup property holds

$$I_+^\alpha I_+^\beta = I_+^{\alpha+\beta}, \quad \Re(\alpha), \Re(\beta) > 0. \tag{14}$$

For $\Re(\alpha) \geq 0$ and $\alpha \notin \mathbb{N}$, the Caputo fractional derivative $D_*^\alpha f$ is defined as [§2] [12]:

$$\mathcal{D}_*^\alpha \{f(t); x\} := I_+^{n-\alpha} \{D^n f(t); x\}, \quad n = [\Re(\alpha)] + 1, \tag{15}$$

where $I_+^{n-\alpha}$ represents the Riemann–Liouville fractional integral defined in Equation (11), and $D^n$ denotes the $n$-th derivative with respect to $x$. For $\alpha \in \mathbb{N}$, the Caputo fractional derivative is given by $\mathcal{D}_*^n = D^n$.

More general fractional operators are the left-side and right-sided Erdélyi–Kober integrals that involve an additional parameter, known as the Erdélyi–Kober parameter. Let $f$ be continuous function on $[0, \infty)$, $\mathrm{Re}(\alpha) > 0$, and $\eta > -\frac{1}{2}$. The left-sided Erdélyi–Kober fractional integral of order $\alpha$ and parameter $\eta$, denoted by $I_{+,2,\eta}^\alpha f(x)$, is given by:

$$(I_{+,2,\eta}^\alpha f)(x) := \frac{2}{\Gamma(\alpha)} \frac{1}{x^{2(\alpha+\eta)}} \int_0^x f(t)(x^2 - t^2)^{\alpha-1} t^{2\eta+1} dt \quad x > 0. \tag{16}$$

Similarly, the right-sided Erdélyi–Kober fractional integral of order $\alpha$ and parameter $\eta$, denoted by $I_{-,2,\eta}^\alpha f(x)$, is defined as:

$$(I_{-,2,\eta}^\alpha f)(x) := \frac{2}{\Gamma(\alpha)} x^{2\eta} \int_x^\infty f(t)(t^2 - x^2)^{\alpha-1} t^{1-2\alpha-2\eta} dt \quad (x \geq 0). \tag{17}$$

To handle even continuous functions $f$ defined on the real line $\mathbb{R}$, we can extend the definition of $I_{\pm,2,\eta}^\alpha f$ by utilizing parity. Specifically, we define $I_{\pm,2,\eta}^\alpha f$ on $\mathbb{R}$ by setting

$$(I_{\pm,2,\eta}^\alpha f)(x) = (I_{\pm,2,\eta}^\alpha f)(|x|) \quad \text{for all } x \in \mathbb{R}.$$

A straightforward computation shows that

$$I_{+,-1/2}^{\alpha+1/2} f(x) = x^{-2\alpha} I_+^{\alpha+1/2} \{ (\frac{f(\sqrt{t})}{\sqrt{t}}, x^2 \}, \tag{18}$$

$$I_+^{\alpha+1/2} f(x) = x^\alpha I_{+,-1/2}^{\alpha+1/2} I\{t f(t^2), \sqrt{x}\}. \tag{19}$$

The following theorem represents the primary result in this section.

**Theorem 3.** *Let $\kappa \geq -1/2$ and $\epsilon > 0$, and the following hold:*

*(i)* $\mathscr{P}_{\kappa+\epsilon} \subset \mathscr{P}_\kappa$;

*(ii)* $f \in \mathscr{P}_\kappa$ *if and only if* $I_{+,2,\kappa}^\epsilon f \in \mathscr{P}_{\kappa+\epsilon}$.

*Furthermore, the mapping $I^{\epsilon}_{+,2,\kappa} : \mathscr{P}_{\kappa} \to \mathscr{P}_{\kappa+\epsilon}$ is one-to-one.*

**Proof.** By utilizing the Sonine integral representation for the Bessel function, as described in [§12.11] [16], we obtain the following result

$$I^{\epsilon}_{+,2,\kappa}\{\mathscr{J}_{\kappa}(u);t\} = \frac{\Gamma(\kappa+1)}{\Gamma(\kappa+\epsilon+1)}\mathscr{J}_{\kappa+\epsilon}(t).$$

Now, let $f \in \mathscr{P}_{\kappa+\epsilon}$, then let there be a non-negative finite Borel measure $\mu$ on $[0, \infty)$ such that

$$f(x) = \int_0^{\infty} \mathscr{J}_{\kappa+\epsilon}(xt)d\mu(t).$$

It follows

$$f(x) = \frac{\Gamma(\kappa+\epsilon+1)}{\Gamma(\kappa+1)} \int_0^{\infty} I^{\epsilon}_{+,2,\kappa}\{\mathscr{J}_{\kappa}(u);t\}d\eta(t)$$

$$= \int_0^{\infty} \mathscr{J}_{\kappa}(xu)d\widetilde{\eta}(u),$$

where

$$d\widetilde{\eta}(u) = \frac{2\Gamma(\kappa+1)}{\Gamma(\kappa+1)\Gamma(\epsilon)} u^{2(\kappa+1)} \int_u^{\infty} t^{-2\kappa+\epsilon}(t^2-u^2)^{\epsilon-1} \, d\eta(t).$$

Therefore, $f \in \mathscr{P}_{\kappa}$. This proves $(i)$ and $(ii)$. To complete the proof, we aim to demonstrate that if $f$ is a bounded measurable function on $[0,\infty)$ and $\alpha, \beta > -1$, such that the integral $\int_0^1 t^{\alpha}(1-t^2)^{\beta}f(xt)dt = 0$, then we can conclude that $f = 0$ almost everywhere. By using the transformation $t \to \sqrt{t}$, we can rewrite the integral as follows:

$$\int_0^1 t^{\alpha}(1-t^2)^{\beta}f(xt)dt = \frac{1}{2}\int_0^1 t^{(\alpha-1)/2}(1-t)^{\beta}f(x\sqrt{t})dt.$$

Applying the Titchmarsh theorem on convolution (see, [2]) to this transformed integral, we can establish the desired result. $\square$

The following theorem, which serves as the second main result in this section, is inspired by the work of R. M. Trigub [§6.3] [2].

**Theorem 4.** *Let $\kappa > -\frac{1}{2}$. In order for $f \in \mathscr{P}_{\kappa}$, it is necessary and sufficient that the function $f(\cdot)$ satisfies:*

(i)  $\left(x^{\kappa}f(\sqrt{x})\right)^{(j)} = 0$, *for $j = 0, 2, \ldots, n$;*
(ii)  $xD\, ^C\mathcal{D}^{\kappa-1/2}_*\{t^{\kappa}f(\sqrt{t}),x^2\} \in \mathscr{P}_{-1/2}$;
*where $\kappa = n+r$ with $-1/2 < r \leq 1/2$.*

**Proof.** Necessity: Suppose $f \in \mathscr{P}_{\kappa}$. By Theorem 3, $f \in \mathscr{P}_{\kappa}$ if and only if there exists $g \in \mathscr{P}_{-1/2}$ such that

$$f(x) = I^{\kappa+1/2}_{+,2,-1/2}\{g(t);x\} = \frac{2}{\Gamma(\kappa+1/2)x^{2\kappa}} \int_0^x g(t)(x^2-t^2)^{\kappa-1/2}dt. \tag{20}$$

Using Equation (18), we can rewrite Equation (20) in the following equivalent form:

$$x^{\kappa}f(\sqrt{x}) = \frac{1}{\Gamma(\kappa+1/2)} \int_0^x g(\sqrt{t})(x-t)^{\kappa-1/2}t^{-1/2}dt.$$

Since the function $t \to \frac{g_0(t)}{\sqrt{t}}$ is locally integrable on $[0, \infty)$, it follows that $x \to x^\kappa f(\sqrt{x})$ belongs to class $C^n$ on $[0, \infty)$, and for $j = 0, 1, \ldots, n$, we have

$$\left[ \frac{d^j}{dx^j} \left( x^\kappa f(\sqrt{x}) \right) \right]_{x=0} = 0,$$

where

$$\kappa = n + r, \quad \text{and} \quad -\frac{1}{2} < r \le \frac{1}{2}.$$

Furthermore,

- for $r = 1/2$, we have

$$g(\sqrt{x}) = \sqrt{x} \left[ x^{n+1/2} f(\sqrt{x}) \right]^{(n+1)} = \mathcal{D}_*^{n+1} \left[ x^{n+1/2} f(\sqrt{x}) \right],$$

- for $-1/2 < r < 1/2$,

$$\frac{d^n}{dx^n} \left[ x^\kappa f(\sqrt{x}) \right] = \frac{1}{\Gamma(r+1/2)} \int_0^x g(\sqrt{t})(x-t)^{r-1/2} t^{-1/2} \, dt. \tag{21}$$

In this case, we can solve the above Abel integral equation to obtain

$$\begin{aligned} g(\sqrt{x}) &= \frac{\sqrt{x}}{\Gamma(\frac{1}{2}-r)} \frac{d}{dx} \int_0^x \frac{d^n}{dt^n} \left( t^\kappa f(\sqrt{t}) \right) \frac{dt}{(x-t)^{r+1/2}}, \\ &= \sqrt{x} D \, {}^C\mathcal{D}_*^{\kappa-1/2} \{ t^\kappa f(\sqrt{t}); x \}. \end{aligned}$$

This proves $(i)$ and $(ii)$.

Sufficiency: Let us assume that $f$ satisfies conditions $(i)$ and $(ii)$. We consider two cases based on the value of $\kappa = n + r$, where $-1/2 < r \le 1/2$.

Case 1: $r = 1/2$ From condition $(ii)$, we have

$$g(\sqrt{x}) = \sqrt{x} \left[ \left( x^{n+1/2} f(\sqrt{x}) \right)^{(n+1)} \right]. \tag{22}$$

Applying $I_+^{n+1}$ to the above equation and using condition $(i)$, we obtain

$$f(x) = \frac{2}{n!} \cdot \frac{1}{x^{2n+1}} \int_0^x (x^2 - t^2)^n g(t) dt \in \mathcal{P}_{n+1/2}. \tag{23}$$

Case 2: $-1/2 < r < 1/2$. From condition $(ii)$, we can write

$$\frac{d^n}{dx^n} \left[ x^\kappa f(\sqrt{x}) \right] = \frac{1}{\Gamma(r+1/2)} \int_0^x g(\sqrt{t})(x-t)^{r-1/2} t^{-1/2} \, dt.$$

Applying $I_+^n$ to the above equation and using condition $(i)$ and semigroup property for the Riemann–Liouville integral in Equation (14), we obtain

$$f(x) = I_{+,2,-1/2}^{\kappa+1/2} \{ g(t); x \} \in \mathcal{P}_\kappa. \tag{24}$$

Therefore, in both cases, we have shown that $f \in \mathcal{P}_\kappa$, which completes the proof of sufficiency. $\square$

## 5. Characterization via Complete Monotone Function

In this section, we explore the characterization of the class $\mathcal{P}_\kappa$ by leveraging Bernstein's theorem for completely monotone functions.

Recall that a function—$f : [0, \infty) \to \mathbb{R}$—is a completely monotone function if it is of class $C^\infty$ on $(0, \infty)$ and

$$(-1)^n f^n(x) \geq 0, \quad x > 0, n \in \mathbb{N}.$$

According to Bernstein's characterization, $f$ is completely a monotone function if and only if there exists some measure $\mu$ on $[0, \infty)$ such that

$$f(x) = \int_0^\infty e^{-tx} d\mu(t).$$

We now focus on the connection between positive, definite radial functions and completely monotone functions, as originally established by Schoenberg in 1938 (see [14]).

**Theorem 5** ([4]). *A function $f$ is completely monotone on $[0, \infty)$ if and only if the function $f(x) := f(\|x\|)$ is positive definite on every $\mathbb{R}^n$.*

The main result in this section is the following theorem.

**Theorem 6.** *For any $\kappa \geq -1/2$, a function $f$ belongs to the class $\mathscr{P}_\kappa$ if and only if $f(\sqrt{x})$ is completely monotone on the interval $[0, \infty)$.*

**Proof.** Sufficiency: By the Bernstein theorem, there exists a finite positive measure $\mu$ on $[0, \infty)$ such that for all $x \geq 0$,

$$f(x) = \int_0^\infty e^{-tx^2} d\mu(t).$$

On the other hand, according to (Formula 4.11.27) [15], we have

$$e^{-tx^2} = \int_0^\infty \mathscr{J}_\kappa(ux) \frac{e^{-\frac{u^2}{4t}}}{(2t)^{\kappa+1}} \sigma\kappa(du). \tag{25}$$

Using this representation, we can rewrite $f(x)$ as

$$f(x) = \int_0^\infty \mathscr{J}_\kappa(tx) d\widetilde{\mu}(t),$$

where

$$d\widetilde{\mu}(t) = \int_0^\infty \frac{e^{-\frac{x^2}{4t}}}{(2t)^{\kappa+1}} d\mu(t) \sigma_\kappa(du).$$

Therefore, we have shown that $f(x) \in \mathcal{P}_\kappa$ for all $\kappa \geq -1/2$. Hence, the sufficiency of the conditions is established.

Necessity: Suppose $f \in \mathcal{P}_\kappa$ for all $\kappa \geq -1/2$. In particular, $f \in \Phi_n$ for all $n \in \mathbb{N}$. Then, the result follows from Theorem 5, which establishes the connection between positive definite radial functions and completely monotone functions. Hence, the necessity of the conditions is demonstrated. $\square$

## 6. Application: Positivity of the Fundamental Solution

In this section, we will show the positivity of the fundamental solution of the following space-fractional Bessel diffusion equation:

$$\begin{cases} \partial_t u(t, x) = -(-\mathscr{B}_\kappa)^{\alpha/2} u(t, x), \\ u(0, x) = f(x), \ x \geq 0, \quad t > 0. \end{cases} \tag{26}$$

Here, $(-\mathscr{B}_\kappa)^{\alpha/2}$ is the fractional Bessel operator, which is given by [19],

$$(-\mathscr{B}_\kappa)^{\alpha/2} f(x) = \frac{2^{\alpha+\kappa} \Gamma(\kappa + \frac{\alpha}{2} + 1)}{\Gamma(\kappa+1) |\Gamma(-\frac{\alpha}{2})|} \int_0^\infty \frac{f(x) - \tau_\kappa^x f(y)}{y^{\alpha+1}} \, dy.$$

Let us denote the Fourier–Bessel transform of a function $u(t, x)$ with respect to $x$ as $\hat{u}(t, \lambda)$, where $\lambda \geq 0$. Applying the Fourier–Bessel transform to both sides of the equation in Equation (26), we obtain:

$$\begin{cases} \partial_t \widehat{u}(t, \lambda) = -\lambda^\alpha \widehat{u}(t, \lambda), \\ \widehat{u}(0, x) = \widehat{f}(\lambda). \end{cases}$$

Then,

$$\widehat{u}(\lambda, t) = \widehat{f}(\lambda) e^{-\lambda^\alpha t}.$$

Therefore,

$$u(t, x) = (\mathscr{G}_t^{\alpha, \kappa} * f)(x),$$

where

$$\mathscr{G}_t^{\alpha, \kappa}(x) = \mathscr{G}^{\alpha, \kappa}(x, t) = \int_0^\infty e^{-\lambda^\alpha t} \mathscr{J}_\kappa(\lambda x) \, \sigma_\kappa(d\lambda). \tag{27}$$

Using the following scaling rules for the Fourier–Bessel transform:

$$\int_0^\infty f(ax) \mathscr{J}_\kappa(\lambda x) \sigma_\kappa(dx) = \frac{1}{a^{2\kappa+2}} \int_0^\infty f(x) \mathscr{J}_\kappa(\lambda x/a) \sigma_\kappa(dx), \quad a > 0,$$

we obtain the following scaling property of the kernel $\mathscr{G}^{\alpha, \kappa}(x, t)$

$$\mathscr{G}^{\alpha, \kappa}(t, x) = t^{-2(\kappa+1)/\alpha} \mathscr{G}^{\alpha, \kappa}(xt^{-1/\alpha}, 1), \quad t > 0, \quad x \in \mathbb{R}.$$

Consequently, introducing the similarity variable $x/t^\alpha$, we can write

$$\mathscr{G}^{\alpha, \kappa}(x, t) = t^{-2(\kappa+1)/\alpha} \mathscr{K}^{\alpha, \kappa}(xt^{-1/\alpha}),$$

where

$$\mathscr{K}^{\alpha, \kappa}(x) = \int_0^\infty e^{-\lambda^\alpha} \mathscr{J}_\kappa(\lambda x) \sigma_\kappa(d\lambda). \tag{28}$$

Particular cases of the density $\mathscr{K}^{\alpha, \kappa}$ are the following [1,8]:

- The density $\mathscr{K}^{2, \kappa}(x)$, $\kappa \geq -1/2$, is the Gaussian density kernel

$$\mathscr{K}^{2, \kappa}(x) = \frac{e^{-\frac{x^2}{4}}}{2^{\kappa+1}}; \tag{29}$$

- The density $\mathscr{K}^{1, \kappa}$, $\kappa \geq -1/2$, is the Poison density

$$\mathscr{K}^{1, \kappa}(x) = \frac{2^{\kappa+1} \Gamma(\kappa + \frac{3}{2})}{\sqrt{\pi}} \frac{1}{(1 + x^2)^{\kappa + \frac{3}{2}}}. \tag{30}$$

**Proposition 2** ([19]). *The following holds:*

1. *For $\kappa \geq -1/2$, $0 < \alpha < 2$, we have*

    - $x \neq 0$

    $$\mathscr{K}^{\alpha, \kappa}(x) = \frac{1}{2^{\kappa+1} \pi} \sum_{n=1}^\infty (-1)^{n+1} \frac{\Gamma(1 + \alpha n/2) \Gamma(\kappa + 1 + \alpha n/2)}{n!} \sin(\frac{\alpha n \pi}{2}) \left( \frac{4}{x^2} \right)^{\frac{\alpha n}{2} + \kappa + 1}.$$

    - $x = 0$, $\mathscr{K}^{\alpha, \kappa}(0) = \frac{\Gamma(\frac{2(\kappa+1)}{\alpha})}{\alpha}$.

2. *For $\kappa \geq -1/2$ and $\alpha > 1$, we have*

$$\mathscr{K}^{\alpha, \kappa}(x) = \frac{1}{\alpha 2^\kappa} \sum_{n=0}^\infty \frac{(-1)^n}{n!} \frac{\Gamma(\frac{2}{\alpha}(n + \kappa + 1))}{\Gamma(\kappa + 1 + n)} \left( \frac{x^2}{4} \right)^n. \tag{31}$$

**Lemma 1.** *Let $v$ be an even function of class $C^\infty$ with compact support in $[-1,1]$, which is positive and satisfies $\|v\|_{1,\kappa} = 1$. For any $\varepsilon \in (0,1]$, define the function $v_\varepsilon(x) = \varepsilon^{-2\kappa-2}v\left(\frac{x}{\varepsilon}\right)$. Then, $v_\varepsilon$ has compact support in $[-\varepsilon,\varepsilon]$ and $\|v_\varepsilon\|_{1,\kappa} = 1$. Furthermore,*

$$|\widehat{v_\varepsilon}(\lambda) - 1| \leq \varepsilon\lambda \quad \text{for all } \lambda \geq 0.$$

**Proof.** First, we note that $v_\varepsilon$ has compact support in $[-\varepsilon,\varepsilon]$ since $v$ has compact support in $[-1,1]$. To show $\|v_\varepsilon\|_{1,\kappa} = 1$, we compute

$$\|v_\varepsilon\|_{1,\kappa} = \int_0^\varepsilon v_\varepsilon(x)\sigma_\kappa(dx) = \int_0^1 v(x)\sigma_\kappa(dx) = \|v\|_{1,\kappa} = 1.$$

It follows

$$\widehat{v_\varepsilon}(\lambda) - 1 = \int_0^\varepsilon v_\varepsilon(x)(\mathscr{J}_\kappa(\lambda x) - 1)\sigma_\kappa(dx)$$

Since

$$|\mathscr{J}_\kappa(x) - 1| \leq x \quad \text{for all } x \geq 0,$$

we have

$$|\widehat{v_\varepsilon}(\lambda) - 1| \leq \varepsilon\lambda \int_0^\varepsilon v_\varepsilon(x)\sigma_\kappa(dx) = \varepsilon\lambda.$$

Therefore, the lemma holds. $\square$

**Proposition 3.** *A continuous bounded function on $[0,\infty)$ is positive definite if and only if for every even function $h$ in the Schwartz space $S(\mathbb{R})$, the inequality $\widehat{h}(x) \geq 0$ for all $x \geq 0$ implies $\widehat{hf}(x) \geq 0$ for all $x \geq 0$.*

For the proof of this proposition, refer to the last remark in [4].

The following theorem represents the main result of this section:

**Theorem 7.** *For $\kappa \geq -1/2$ and $0 < \alpha \leq 2$, we have $\mathscr{G}^{\alpha,\kappa}(x,t) \geq 0$.*

**Proof.** Since the function $e^{-\lambda^\alpha}$ is completely monotone for $0 < \alpha < 2$, it follows from Theorem 6 that it is positive definite for all $\kappa \geq -1/2$. Using the notation of Lemma 1, we have

$$v_\varepsilon(x) = \int_0^\infty \widehat{v_\varepsilon}(\lambda)\sigma_\kappa(d\lambda).$$

Since $\widehat{v_\varepsilon}(\lambda) \in S(\mathbb{R})$ is even and positive, and the function $e^{-\lambda^\alpha}$ is positive definite according to the above proposition, we have

$$\int_0^\infty \widehat{v_\varepsilon}(\lambda)e^{-\lambda^\alpha}\mathscr{J}_\kappa(\lambda x)\sigma_\kappa(d\lambda) \geq 0.$$

Lemma 1 shows that this integral converges to $\mathscr{K}_t^{\alpha,\kappa}(x)$, as $\varepsilon \to 0$. This completes the proof of the theorem. $\square$

## 7. Concluding Remarks

In conclusion, this study has made significant contributions to the field by investigating the continuation and characterization of radial positive definite functions beyond integer dimensions. By utilizing fractional derivatives and exploring their relationship with non-integer continuation, a deeper understanding of these functions has been achieved.

The practical relevance of the findings has been demonstrated through the investigation of the space fractional diffusion equation. By applying the obtained characterizations,

the positivity of the fundamental solution has been established. This showcases the practical implications and real-world applications of continued radial positive definite functions.

Overall, this research expands the knowledge and understanding of radial positive definite functions, paving the way for further advancements in the field. The insights gained from this study have the potential to impact various mathematical disciplines, as well as practical domains such as image processing, data analysis, and machine learning. By extending the study beyond integer dimensions, new avenues for research and applications open, contributing to the advancement of the field as a whole.

**Funding:** This research was funded by the Deputyship for Research and Innovation, Ministry of Education, in Saudi Arabia grant number IFKSUOR3-368-1.

**Institutional Review Board Statement:** Not applicable.

**Informed Consent Statement:** Not applicable.

**Data Availability Statement:** Not applicable.

**Acknowledgments:** The author extends his appreciation to the Deputyship for Research and Innovation, Ministry of Education, in Saudi Arabia.

**Conflicts of Interest:** The author declares no conflict of interest.

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
