# Peer review of "Continuation of Radial Positive Definite Functions and Their Characterization"

_fractalfract, doi:10.3390/fractalfract7080623_

Round 1

Reviewer 1 Report

Please make the following correction:

1. Provide the abstract part of the article in detail.

2. On page number 5, why it is [?]? Correct it .

3. Polish your English language.

4. Make the introduction section more strong by adding some relevant references, especially in the field of Fractional calculus.

Polish your English language and grammar.

Author Response

I hope this email finds you well. I'm writing to submit a revised version of the paper entitled “Continuation of Radial Positive Definite Functions. I have taken into consideration the recommendations and remarks provided by the reviewers and have made significant improvements to my manuscript.

Sincerely yours

Reviewer 2 Report

Regarding the manuscript entitled “Continuation of radical positive definite functions and their characterization”,

by Fethi Bouzeffour, which is submitted for publication in “Fractal and Fractional”.

The author studies radical positive definite functions and investigates the link between the aforementioned concept

and the ones of fractional integral/derivative as well as of completely monotonic functions. At the end, he gives an

application of the above results, where the non-negativity of the fundamental solution of a certain fractional PDE is

demonstrated.

The manuscript is publishable, but the following minor revision is advised.

ˆ p.3: Replace Φ(R) with Φ(Rn).

ˆ p.4: Why f0 in the definition of radial functions has to be even? E.g, why would not the identity function (odd)

work? Besides, we do not care about the behaviour of the functions for negative values, since the Euclidean

norm is always non-negative.

ˆ p.5: At the end of the proof of Proposition 3.7, a reference is written as [?].

ˆ p.6: In equation (4.3) the gamma function is defined, but this function has already been employed for the first

time in p.2.

ˆ p.6: Replace Erdelyi with Erd´elyi.

ˆ p.9: The concept of completely monotonicity concerns smooth functions. Such fact is not taken into account in

both Theorem 5.1 and Theorem 5.2.

ˆ p.12-13: References [1], [3] and [28] need to be corrected.

1

Author Response

(The authors gave the same response as above.)

Reviewer 3 Report

This paper is very interesting in fractional theory but needs minor revision.

1.      The authors should give the novel of this considered model in abstract.

2.      The introduction is not well organized. The authors should give the developing process of the non-integer orders, or fractional derivatives and the aims.

3.      The author pays attention to some grammar and spelling errors in this article.

4.      The authors should give the sources of Definition 3.1 and Theorem 3.2/3.5.

5.      Missing references in the previous paragraph of Part Four.

6.      The authors should analyze this solution (6.5).

7.      The authors should give the difference about R-L fractional derivative and Caputo fractional derivative from these results.

8.      Labels not used by the author in the paper should be deleted.

In order to enrich the introduction and attract more readers, some important works related with recent development in soliton theory theory and its applications should be discussed in the introduction part and be added in the references lists: Numerical Methods for Partial Differential Equations,35.4 (2019): 1305-1325; International Journal of Geometric Methods in Modern Physics, 19(11) (2022) 2250173, Fractals, 31(5) (2023) 2350033, General Fractional Derivatives: Theory, Methods and Applications, CRC Press, New York, USA, 2019.

I recommend it for publication after minor revision.

This paper is very interesting in fractional theory but needs minor revision.

1.      The authors should give the novel of this considered model in abstract.

2.      The introduction is not well organized. The authors should give the developing process of the non-integer orders, or fractional derivatives and the aims.

3.      The author pays attention to some grammar and spelling errors in this article.

4.      The authors should give the sources of Definition 3.1 and Theorem 3.2/3.5.

5.      Missing references in the previous paragraph of Part Four.

6.      The authors should analyze this solution (6.5).

7.      The authors should give the difference about R-L fractional derivative and Caputo fractional derivative from these results.

8.      Labels not used by the author in the paper should be deleted.

In order to enrich the introduction and attract more readers, some important works related with recent development in soliton theory theory and its applications should be discussed in the introduction part and be added in the references lists: Numerical Methods for Partial Differential Equations,35.4 (2019): 1305-1325; International Journal of Geometric Methods in Modern Physics, 19(11) (2022) 2250173, Fractals, 31(5) (2023) 2350033, General Fractional Derivatives: Theory, Methods and Applications, CRC Press, New York, USA, 2019.

I recommend it for publication after minor revision.

Author Response

(The authors gave the same response as above.)

Round 2

Reviewer 1 Report

NA